# VO$_2$ memristor-based frequency converter with in-situ synthesize and mix for wireless internet-of-things

Chang Liu[1], Pek Jun Tiw[1], Teng Zhang [1], Yanghao Wang [1], Lei Cai[1], Rui Yuan[1], Zelun Pan [1], Wenshuo Yue[1], Yaoyu Tao [1,2] ✉ & Yuchao Yang [1,2,3,4] ✉

Wireless internet-of-things (WIoT) with data acquisition sensors are evolving rapidly and the demand for transmission efficiency is growing rapidly. Frequency converter that synthesizes signals at different frequencies and mixes them with sensor datastreams is a key component for efficient wireless transmission. However, existing frequency converters employ separate synthesize and mix circuits with complex digital and analog circuits using complementary metal-oxide semiconductor (CMOS) devices, naturally incurring excessive latency and energy consumption. Here we report a highly uniform and calibratable VO$_2$ memristor oscillator, based on which we build memristor-based frequency converter using 8 × 8 VO$_2$ array that can realize in-situ frequency synthesize and mix with help of compact periphery circuits. We investigate the self-oscillation based on negative differential resistance of VO$_2$ memristors and the programmability with different driving currents and calibration resistances, demonstrating capabilities of such frequency converter for in-situ frequency synthesize and mix for 2 ~ 8 channels with frequencies up to 48 kHz for low frequency transmission link. When transmitting classical sensor data (acoustic, vision and spatial) in an end-to-end WIoT experimental setup, our VO$_2$-based memristive frequency converter presents up to 1.45× ~ 1.94× power enhancement with only 0.02 ~ 0.21 dB performance degradations compared with conventional CMOS-based frequency converter. This work highlights the potential in solving frequency converter's speed and energy efficiency problems in WIoT using high crystalline quality epitaxially grown VO$_2$ and calibratable VO$_2$-based oscillator array, revealing a promising direction for next-generation WIoT system design.

Wireless internet of things (WIoT) networks are becoming increasingly attractive in a wide range of applications such as smart homes[1–10], industrial automation[11–14] and medical monitoring[15,16]. WIoT hardware can communicate with each other or with central cloud without cables or physical connections, enabling easy setup compared to traditional network and reducing the need for manual intervention. State-of-the-art WIoT hardware[1–16] used in smart homes or smart factories need to transmit data at low frequency bands (down to tens of kHz) for long-

[1]Beijing Advanced Innovation Center for Integrated Circuits, School of Integrated Circuits, Peking University, Beijing 100871, China. [2]Center for Brain Inspired Chips, Institute for Artificial Intelligence, Frontiers Science Center for Nano-optoelectronics, Peking University, Beijing 100871, China. [3]School of Electronic and Computer Engineering, Peking University, Shenzhen 518055, China. [4]Center for Brain Inspired Intelligence, Chinese Institute for Brain Research (CIBR), Beijing, Beijing 102206, China. ✉e-mail: taoyaoyutyy@pku.edu.cn; yuchaoyang@pku.edu.cn

range communication with central clouds and energy efficient decision-making as well as action. These help to improve productivity, reduce costs and enhance customer satisfaction. A typical wireless data transmission link for WIoT networks consists of four modules: (1) data acquisition sensors which typically sense vision, acoustic or spatial signals, etc.; (2) pre-processor hardware that generates datastreams to be transmitted based on sensor data; (3) frequency converter that synthesizes signals at different frequencies based on target frequency bands and mixes them with datastreams to produce output signals; and (4) radio frequency modules such as filter, attenuator, power amplifier and antenna to send data via electromagnetic waves. Among them, frequency converter[12,17–22] is a key block to enable energy efficient wireless data transmission, which is used to create new frequencies based on band requirements and mixing them with sensor datastreams for improved data transmission efficiency. In its most common applications, multiple signals with desired frequencies can be generated and the frequency converter produces new signal at the sum of the original frequencies.

State-of-the-art frequency converter designs usually employ complementary metal-oxide semiconductor (CMOS) based digital and analog circuits[20–22] with separate frequency synthesizer and mixer circuitry using registers, oscillators, digital-to-analog converter (DAC) and operational amplifiers, supporting unnecessarily high frequencies (up to GHz) for WIoT that naturally incurring excessive latency and power. Frequency control registers are used to control the frequencies generated at multiple numerically controlled oscillators, after which a group of DACs are used to convert the digital samples into analog waves. The analog waves themselves are then mixed together with data waves to be transmitted using CMOS mixers such as operational amplifier-based circuits to produce the output signals with target summed frequencies. The system performance using such complex circuits are becoming increasingly challenging to meet tighter constraints when constructing next-generation WIoT network[23–27] at low frequency bands. To overcome these problems, a frequency converter supporting up to tens of kHz frequency bands that can realize frequency synthesize and mix functions within a single module for improved latency and energy consumption is greatly desired for WIoT hardware.

To realize this goal, memristors are especially attractive because of their rich ion dynamics[28–40] and electrical behaviors that support tunable oscillation frequencies. Although the hardware implementations of oscillatory bionic neurons are widely reported[41–43], there is no literature that utilizes the oscillation characteristics to build frequency converter. Moreover, calibratable array based on memristor oscillators can realize frequency synthesize and mix functions in a single module with help of compact circuits, which has great potential to improve latency and energy consumption of WIoT hardware. More importantly, an end-to-end WIoT experimental setup based on such frequency converter has not been reported to date.

In this article, we report a frequency converter based on epitaxial $VO_2$ memristor grown by pulsed laser deposition. Inspired by $VO_2$ memristor's capability of synthesizing signals at different frequencies, we propose $VO_2$ memristor array design with compact calibration circuitries to extend its frequency synthesize capability to frequency mix. $VO_2$ memristor exhibits self-oscillation phenomenon based on negative differential resistance (NDR) behavior shown in the current-driven *I-V* curves. Recorded optically visible filaments show that self-oscillation within the NDR regions is due to the formation and disappearance of a high temperature conducting channel, which is a result of electrothermally-induced Mott transition. Thus, a highly uniform and calibratable oscillator based on $VO_2$ memristor array has been experimentally implemented for the first time. The oscillator exhibits excellent cycle-to-cycle and device-to-device uniformity, due to the high crystalline quality of epitaxially grown $VO_2$ and the introduction of parallel calibration resistor. With help of voltage-controlled current drivers and calibration resistors, our frequency converter

system is able to in-situ synthesize and mix for 2 ~ 8 channels with frequencies up to 48 kHz. The synthesized and mixed frequencies can be programed by the $VO_2$ array size, the voltage-controlled current mirror circuitries and the calibration resistances. We conduct an in-depth analysis of the programmability of the $VO_2$ array-based oscillators. Using a one-time phase matcher with delay-line based phase matcher to align phases, we experimentally demonstrate our frequency converter in three representative real-world WIoT applications (vision, acoustic and spatial data transmission) with an end-to-end software-hardware co-designed experimental setup. Results show that our frequency converter outperforms conventional CMOS-based designs by 1.45× ~ 1.94× in power with only 0.02 dB–0.21 dB BER degradations, respectively, when transmitting acoustic, vision and point cloud data. The in-situ synthesize and mix functions within a single module has not been reported to date. This concept, design work and experimental results using $VO_2$-based memristors reveal a prospect to construct highly energy efficient and programmable frequency converter, providing a promising direction to the developments of next-generation WIoT system.

## Results

### In-situ frequency synthesize and mix in WIoT system

As shown in Fig. 1a, in classical WIoT applications such as smart homes or smart factories, a variety of sensors are utilized in different kinds of WIoT hardware, including audio sensors for acoustic signals, vision sensors for image signals and depth sensors for point cloud signals. WIoT hardware collect these physical information and interact with other WIoT hardware in short ranges at high frequency bands up to GHz level, or communicate with remote central cloud in long ranges at low frequency bands down to tens of kHz. Sensor data are first pre-processed into datastreams for wireless transmission as shown in Fig. 1b. However, these raw datastreams cannot be directly used for radio frequency modules and a frequency converter is needed to synthesize signals at different frequencies according to transmission bands and mix them with the raw datastreams. State-of-the-art frequency converter designs rely on CMOS-based digital and analog circuits. These designs usually consist of two parts, one for frequency synthesize and the other for frequency mix. Figure 1c shows a classical CMOS-based frequency converter using separate synthesizer and mixer circuitries, where frequency synthesizer is built based on frequency control registers, parallel oscillators and digital-to-analog converters (ADCs) and frequency mixer is typically designed using operational amplifiers as well as resistances or capacitances. Realizing a frequency converter that can carry out synthesize and mix simultaneously within a single module for improved frequency conversion efficiency is desired but not yet available. To achieve this goal, our $VO_2$-based frequency converter aims for in-situ synthesize and mix, as shown in Fig. 1d to enable a highly efficient WIoT system.

### $VO_2$ memristor-based calibratable oscillator

We start by developing a $VO_2$ memristor-based oscillator, which is the elementary block for synthesizing signals at different frequencies. To efficiently construct oscillators, $VO_2$ memristors with highly uniform threshold switching[42–44] and volatile properties are required. As illustrated in Fig. 2a, the $VO_2$ memristor developed in this work consists of two Au/Ti electrodes sandwiching a $VO_2$ film in a planar device. Supplementary Fig. 3a shows a scanning electron microscope (SEM) image of the device, and the channel region is enlarged in supplementary Fig. 3b for details, where the channel length is 5 μm and the electrode width is 5 μm (see Methods for fabrication process details). As shown in Supplementary Fig. 1a, the quality of the $VO_2$ film was confirmed electrically through isothermal electrical resistance characterization of the metal-insulator transition (MIT) in lateral device. The MIT shows repeatable resistance switching with a magnitude of ≈400×. Supplementary Fig. 1b further confirms the critical temperatures of about

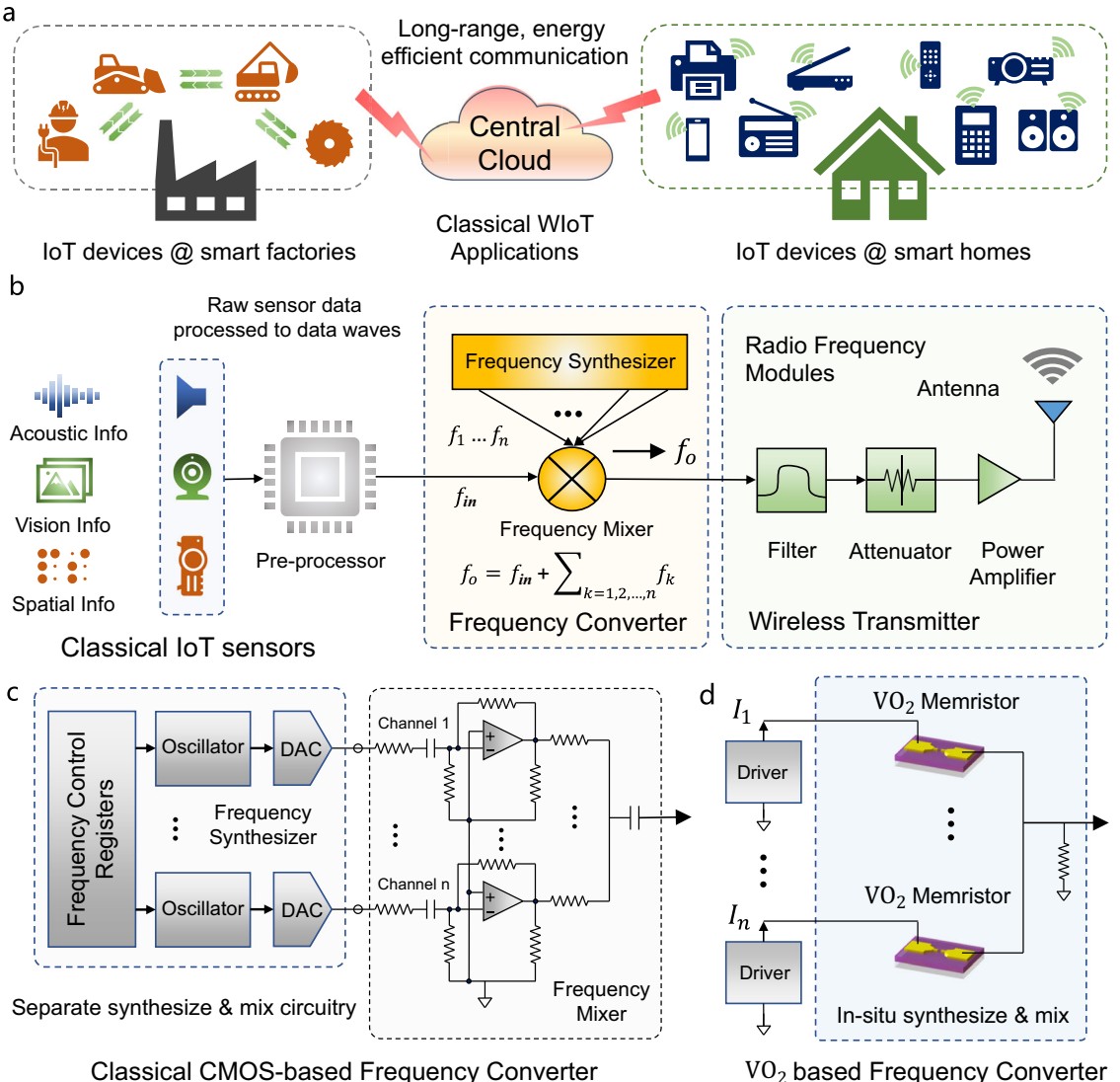

**Fig. 1 | Frequency converter in wireless internet-of-things (WIoT). a** Classical wireless IoT applications at smart home or smart factory interacting with remote central cloud at low frequency bands. **b** Wireless data transmission link in WIoT hardware, including a variety of sensors, pre-processor, frequency converter and radio frequency circuitries. **c** Classical CMOS-based frequency converter using complex digital and analog circuits with separate frequency synthesizer and mixer modules, incurring excessive latency and energy consumption. **d** Our VO$_2$-based memristive oscillators build a frequency converter with in-situ frequency synthesize and mix for reduced latency and energy.

345 K and 339 K for the heating and cooling processes, respectively. The large changes in resistance during switching and the measured thermal hysteresis are similar to the experimental values found in the literature for undoped VO$_2$[45]. The VO$_2$ crystal quality were also characterized by X-ray diffraction (XRD), Raman spectroscopy, X-ray photoelectron spectroscopy (XPS), and scanning transmission electron microscope (STEM). The results of the XRD taken both before and after the VO$_2$ film was deposited are depicted in Supplementary Fig. 1c. According to the findings of the XRD analysis, there is a single diffraction peak at 39.8 degrees occurring in the film, which represents (020) monoclinic VO$_2$ films formed on $c$-plane Al$_2$O$_3$ substrates[46]. The Raman measurements (Supplementary Fig. 1d) showed that all the peaks are well aligned with reported monoclinic VO$_2$ results[47] in the literature. The chemical composition and valence states of the VO$_2$ microcrystals were investigated using XPS. The confirmation of the presence of elements C, O, and V was achieved through the observation of their respective XPS characteristic peaks, as depicted in Supplementary Fig. 2a. The observed peaks for element C may potentially be attributed to the presence of adventitious carbon on the surface of the samples, as depicted in Supplementary Fig. 2b. The O 1$s$ peak (Supplementary Fig. 2c) can be fitted with three peaks at 530.50 (V-O bond), 532 eV (O-H bond resulting from physical absorption of H$_2$O on the surface of the sample) and 532.67 eV (C-OH bond originating from organic hydroxyl) that all correspond to the oxygen valence of O$^{2-}$. The high-resolution XPS spectrum of the V 2$p^{3/2}$ (Supplementary Fig. 2d) shows that clear peaks can be deconvoluted into V$^{4+}$ and V$^{5+}$ states centered at 516.21 eV and 517.35 eV, respectively[48]. Only the first 1–3 nm of the sample volume are detected by XPS, which means that surface oxidation is primarily responsible for the large shoulder linked to V$^{5+}$. Supplementary Fig. 3c, d shows the transmission electron microscope (TEM) image of the device, and Fig. 1b, c shows the magnified view of the VO$_2$ film as well as the corresponding fast Fourier transform. The ordered lattice fringes are very clear, verifying that the VO$_2$ has a high crystalline quality with a monoclinic phase, which is very important to achieve high uniformity. Supplementary Fig. 4 shows a cross-sectional STEM image of the device, the energy dispersive X-ray spectroscopy

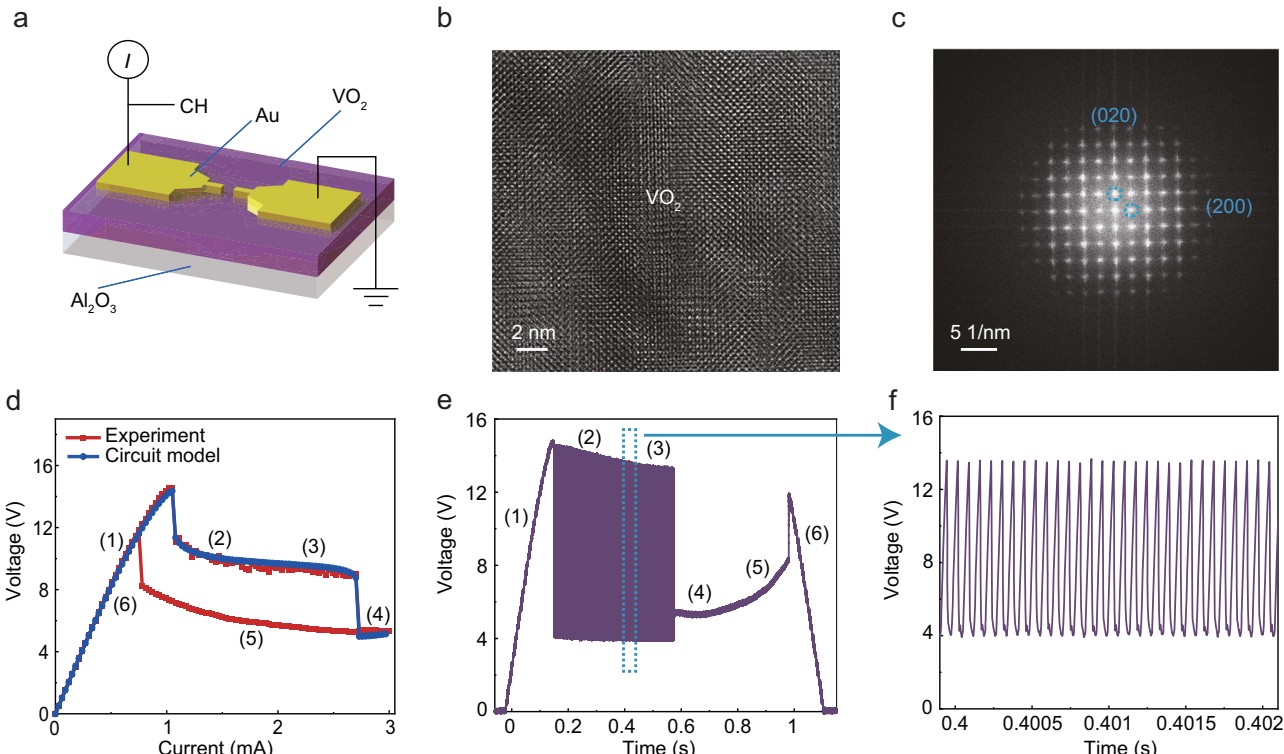

**Fig. 2 | Characteristics of Au/VO₂/Au memristor. a** Schematic of the Au/VO₂/Au planar structure with electrical characterization setup. **b** High-resolution TEM image of the VO₂ region. **c** The diffraction pattern extracted by fast Fourier transformation of (**b**). **d** Current-driven current–Voltage (*I-V*) characteristics of the memristor and simulated *I-V* curves from circuit model. **e** Voltage oscillations when measuring the current-driven *I-V* characteristics of the device. **f** Zoom-in views of the green box.

(EDS) mapping of O, Al, Ti, V and Au elements within the same area and a representative EDS element line profile.

Such a VO₂ memristor exhibits a characteristic current-controlled negative differential resistance (NDR) behavior as shown in the current-driven *I-V* curves in Fig. 2d. The memristor displayed a gradual NDR within a certain current biasing range (~1 mA to 2.7 mA), as well as an abrupt NDR at ~2.7 mA, which formed a hysteresis upon sweeping backward to 0 A[48]. The *I-V* curves with 100 cycles are shown in Supplementary Fig. 5a, demonstrating extremely stable NDR characteristics with low cycle-to-cycle (C2C) variation. While measuring the current-driven *I-V* curves, the voltage across the memristor was read out using an oscilloscope (Fig. 2e). It can be seen from the enlarged picture in Fig. 2f that in the state 2 and 3 regions, the device has self-oscillation phenomenon. This is due to the fact that the quasi-static NDR is an intrinsic characteristic of local activity and, under proper conditions, can lead to electrical (and thermal) self-oscillations[48]. However, the NDR regions were unstable under a voltage-driven *I-V* curves, as shown in Supplementary Fig. 5b, thus resulting in a typical threshold switching behavior along with a hysteresis. Optically visible filaments have been reported in similar planar VO₂ devices and are attributed to local temperature rise within the filament[49]. As evident in Supplementary Movie 1 (The circuit used to obtain the optical micrograph movie is shown in Fig. 3a, with the parallel capacitance and current bias being 110 μF and 1.6 mA, respectively. The lower panel in the movie is the optical micrograph, and the upper panel is the generated spike signal $V_{mem.}$), the resistive switching phenomena within the NDR regions is due to the formation and disappearance of a high temperature conducting channel, which is a result of electrothermally-induced Mott transition[48-51]. For oscillations to emerge when a current source is connected to the VO₂ memristor, two circuit conditions must be met. First, the memristor must be biased such that the load line intersects the NDR region and stabilizes the locally-active state at the

intersection[50,52]. This is rather easy to achieve when a current source is used to drive the circuit as a typical current source has an extremely large parallel resistance and, thus, a load line that is almost parallel to the voltage axis. The second condition is given by linear circuit analysis at the local scale, where a sufficiently large parallel capacitive coupling must be present to destabilize the locally-active bias state[48,53]. It is worth elaborating that when a voltage source is used and analyses are made based on the threshold switching DC characteristic instead, the voltage source and the series source resistance must be chosen such that the load line intersects both the abrupt edges and neither of the two stable *I-V* branches, resulting in no stable circuit states. As the NDR regions fit within the hysteresis in the voltage sweep *I-V* curve, the biasing conditions when using a voltage or current source is used are essentially equivalent. Using a SPICE-based electro-thermal compact model (Supplementary Fig. 6, Supplementary Table 1, Supplementary Note 1), we simulated the *I-V* characteristics of the VO₂ memristor. As shown in Fig. 2d, the simulated curve depicted by the blue line fits very well with the experimental data.

Since the current-driven *I-V* characteristic can generate self-oscillation without an external resistance, the oscillation behavior of oscillator can be realized by a simple circuit, and its configuration is shown in Fig. 3a. The epitaxial VO₂ memristor is connected in parallel with a capacitor. In addition, a 25 Ω resistor $R_0$ is also used to convert the current to a voltage output. The oscilloscope is used to measure electrical waveforms across the VO₂ memristor and the spiking signal of the resistor $R_0$ through channels 1 and 2, respectively (see methods and Supplementary Fig. 7). The intrinsic capacitance of the VO₂ memristor provides the dynamics for integration. When a current bias within the NDR regions is applied to the oscillator, the oscillations characteristic can be completely read from the oscilloscope, and were highly periodic (Supplementary Figs. 8, 9). The spiking rate of the oscillator is largely dependent on applied current bias and parallel

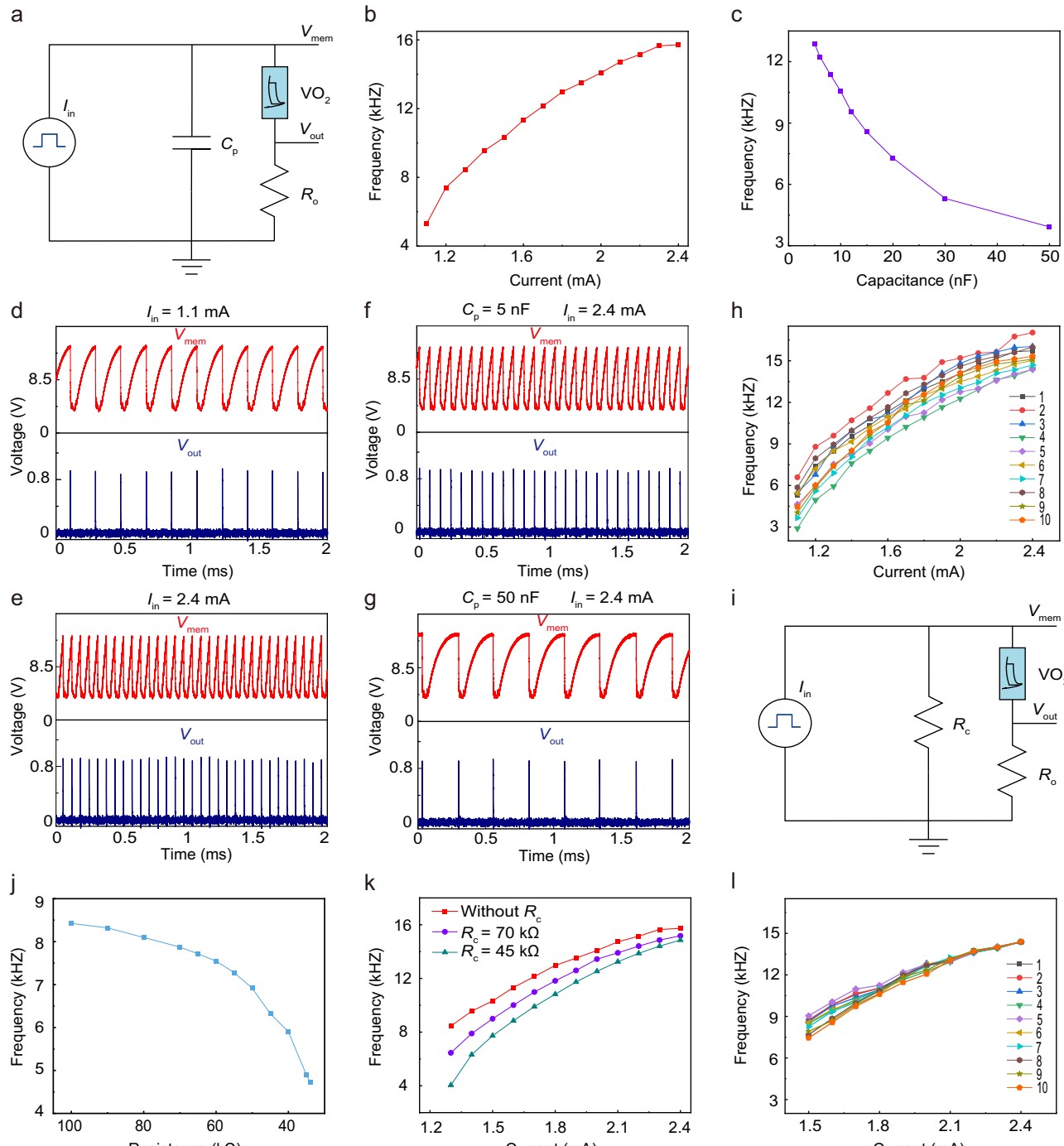

**Fig. 3 | Oscillator and its calibration design circuit. a** Schematic diagram of the oscillators based on VO$_2$ device. **b** The effect of different applied current on oscillation frequency. **c** The effect of the different parallel capacitor on oscillation frequency. **d–g** The oscillatory response under different applied current and parallel capacitor. **h** The effect of applied current ($I_{in}$) on oscillation frequency of different oscillators. Variation from oscillator to oscillator is easily observed. **i** Schematic diagram of the circuit structure of calibratable oscillator. **j** The effect of the calibratable resistance $R_c$ on oscillation frequency. **k** The effect of different applied current on oscillation frequency when using different $R_c$. **l** Oscillation frequency versus applied current for different oscillators with calibrated resistance $R_c$. It is clear that the variation between oscillators is effectively reduced compared to (**h**).

capacitance. Figure 3d,e shows the oscillating behavior of the oscillator under different applied current bias $I_{in}$ (1.1 mA, 2.4 mA) without an external parallel capacitor, more results with >10 different current bias $I_{in}$ values can be found in Supplementary Fig. 8. A larger input current will speed up the charging process, thus increasing the firing frequency (Fig. 2b). Figure 3c, f, g and Supplementary Fig. 9 shows the experimental response of the oscillator under different parallel capacitors when the applied current bias is fixed at 2.4 mA. A larger parallel

capacitance results in a slower integration process and, therefore, a lower firing frequency. The results of the SPICE-based electro-thermal compact model based on VO$_2$ memristor showed excellent agreement with the experimental results (Supplementary Fig. 10). Significantly, the VO$_2$ oscillator exhibited a remarkable endurance of >10$^6$ switching cycles, as depicted in Supplementary Fig. 11. This characteristic guarantees the dependability of our frequency converter that incorporate these particular devices. The temperature may vary depending on the

setting in wireless Internet of Things applications. Therefore, determining the correlation between the frequency of the $VO_2$ oscillator and temperature is crucial. During the temperature-elevated test, only the $VO_2$ device was heated in the probe station, while other components were kept at room temperature. The experimental results (Supplementary Fig. 12) demonstrate that the frequency of the $VO_2$ oscillator rises with increasing temperature. This should be taken into account using temperature-dependent oscillation models such as the one described in ref. 41. when deploying our frequency converter in real-world applications.

The high crystalline quality of epitaxial $VO_2$ has led to low cycle-to-cycle variations (as shown in Figs. 2, 3 and Supplementary Figs. 6, 7). While device-to-device (D2D) variations across the entire chip area still exist (Supplementary Fig. 13a), this indicates that some nominal differences in material and device performance over the entire $1\,cm^2$ chip area may be due to deviations in element composition or effective device volume, etc. Therefore, we have tested the modulation curves of 10 oscillators under current bias without an external parallel capacitor as shown in Fig. 3h and Supplementary Fig. 13b–k. Despite the similarity in modulation trends between oscillators, there are still considerable variations between them, posing significant challenges for application. In order to further reduce the D2D variation, we introduced a parallel calibration resistor $R_c$ into the oscillator circuit (Fig. 3i), and the principle of parallel resistor shunt can effectively improve the D2D variations. Different calibration resistor $R_c$ can effectively change the spiking frequency of the oscillator in a wide range when a bias current is fixed at 1.4 mA, as demonstrated in Fig. 3j and Supplementary Fig. 14. In addition, Fig. 3k shows that the $f$–$I_{in}$ curve can be well controlled by the calibration resistor $R_c$ (more experimental data are demonstrated in Supplementary Figs. 15, 16). Thus, this provides a valuable regulatory means to transfer and tune all modulatory features from different oscillators. In fact, the experimental results show that D2D changes (Fig. 3l) have been effectively reduced compared to Fig. 3h. Thus, the combination of epitaxial $VO_2$ and calibration resistors resolved the C2C and D2D variations, respectively, thereby significantly enhancing the uniformity of oscillators.

### $VO_2$ memristor-based frequency converter with in-situ synthesize and mix

Based on the oscillator introduced in the previous section, we design a frequency converter with in-situ synthesize and mix utilizing a programmable $VO_2$ memristor array. Supplementary Fig. 17 shows the optical image of the $8 \times 8$ array, which consists of eight $8 \times 1$ arrays, a single electrode is directly connected to the current source, and the other electrodes are connected to each other. We designed a circuit schematic for same frequency accumulation using $VO_2$ memristor arrays as frequency converter, as shown in Fig. 4a. Supplementary Fig. 18 presents the experimental results of an oscillator when only one current bias (1.41 mA and 1.42 mA, respectively) was applied. However, when two bias currents are applied at the same time, the resistive voltage pulse signal $V_{out}$ read on the electrodes connected to each other can be mutually accumulated (as demonstrated in Fig. 4b), therefore, it can be well used as a frequency converter. According to the experimental results in Fig. 4c, it can be clearly seen that using two bias current sources (1.41 mA and 1.42 mA) at the same time will generate a $V_{out}$ signal (third panel) equivalent to the sum of the signals generated by the two bias currents alone (first panel and second panel). The frequency converter we designed not only realizes the accumulation of the same frequency, but also realizes the accumulation of different frequencies (as shown in Fig. 4d). Supplementary Fig. 19 shows the experimental results for different frequencies when only one current bias is applied (1.5 mA and 1.22 mA, respectively). When these two bias currents are applied simultaneously, the voltage pulse signal $V_{out}$ read from the electrodes connected to each other can

be mutually accumulated, as shown in Fig. 4e. The experimental results in Fig. 4f clearly shows that this resulting $V_{out}$ signal (third panel) is equivalent to the sum of each $V_{out}$ signal (1st panel and 2nd panel) obtained by applying the bias currents individually. Supplementary Fig. 20 depicts current-driven sweep from 0 to −3 mA, demonstrating oscillating characteristics. Given such a result, this frequency converter can also implement frequency subtraction (Supplementary Fig. 21). Supplementary Fig. 21b, c shows the experimental results when only one positive or negative current bias (1.8 mA and −1.19 mA, respectively) was applied. When the positive and negative bias currents are applied simultaneously, the $V_{out}$ pulse signal can undergo subtraction, as shown in Supplementary Fig. 21d. This is also evident from the experimental results displayed in Supplementary Fig. 21e, wherein the effective $V_{out}$ signal (third panel) is equivalent to the sum of the $V_{out}$ signals corresponding to each bias current (first panel and second panel). Not only that, the frequency converter of this design can also realize the accumulation of 8 frequencies through 8 oscillators, as shown in Fig. 4g. Supplementary Fig. 22 presents the experimental results of an oscillator when only one current bias (1.2 mA, 1.18 mA, 1.19 mA, 1.17 mA, 1.21 mA, 1.17 mA, 1.18 mA and 1.17 mA, respectively) was applied. When 8 bias currents are applied simultaneously, the voltage pulse signals $V_{out}$ generated on the resistor $R_o$ can be mutually accumulated (as shown in Fig. 4h). According to the experimental results in Fig. 4i, it can be clearly seen that using 8 bias current sources simultaneously will generate a $V_{out}$ signal (ninth panel), which is equivalent to the summation of the $V_{out}$ signals generated by each of the 8 bias currents alone (first panel to eighth panel).

### WIoT experiments with $VO_2$ memristor-based frequency converter

After demonstrating the in-situ frequency synthesize and mix capabilities of proposed $VO_2$-based memristive frequency converter for 2 ~ 8 channels with related calibration methodologies, here we build an end-to-end WIoT experimental setup based on such frequency converter for classical WIoT data transmission tasks of acoustic, vision and point cloud data to showcase its performance in real-world applications.

The hardware and software co-designed experimental setup starts from software where sensor data (vision, acoustic or spatial) are preprocessed with encoder and modulator. The encoder we used in this setup is built based on standard Huffman source encoding and rate-1/2 convolutional encoding as shown in Fig. 5a. The resulting datastreams are then sent to modulation block, where binary phase shift keying (BPSK) is used to map the bit 0 to +1 signal amplitude and bit 1 to −1 signal amplitude. Using experimental data collected in previous section when testing the $VO_2$ memristor-based frequency converter, we map target frequencies specified by the transmission bands to control voltages that drive the current source circuitries on the testing PCB as shown in Fig. 5b. Detailed dataflow between each block and the interfaces between software and testing PCB are given in Supplementary Figs. 27, 28. Note that the signal waves generated by our memristive frequency converter are not exactly sinusoidal and can potentially introduce phase misalignment, which may affect bit error rates of data transmission. Additionally, we use delay line-based phase matcher to align the phases of the output signal from the memristive frequency converter to fulfill the requirements of the follow-up radio frequency module. The mixed signals are filtered, attenuated, power amplified and transmitted through additive white Gaussian noise (AWGN) channel with varying signal-to-noise ratio (SNR). Upon receiving the signals from the channel, demodulation and decoding recover corrupted bits (caused by channel noise) in the received datastreams, based on which the sensor data are reconstructed at the receiver side (e.g., central cloud in WIoT).

We experimentally test our $VO_2$-based FC in transmitting three representative sensor signals with vision, acoustic and spatial data.

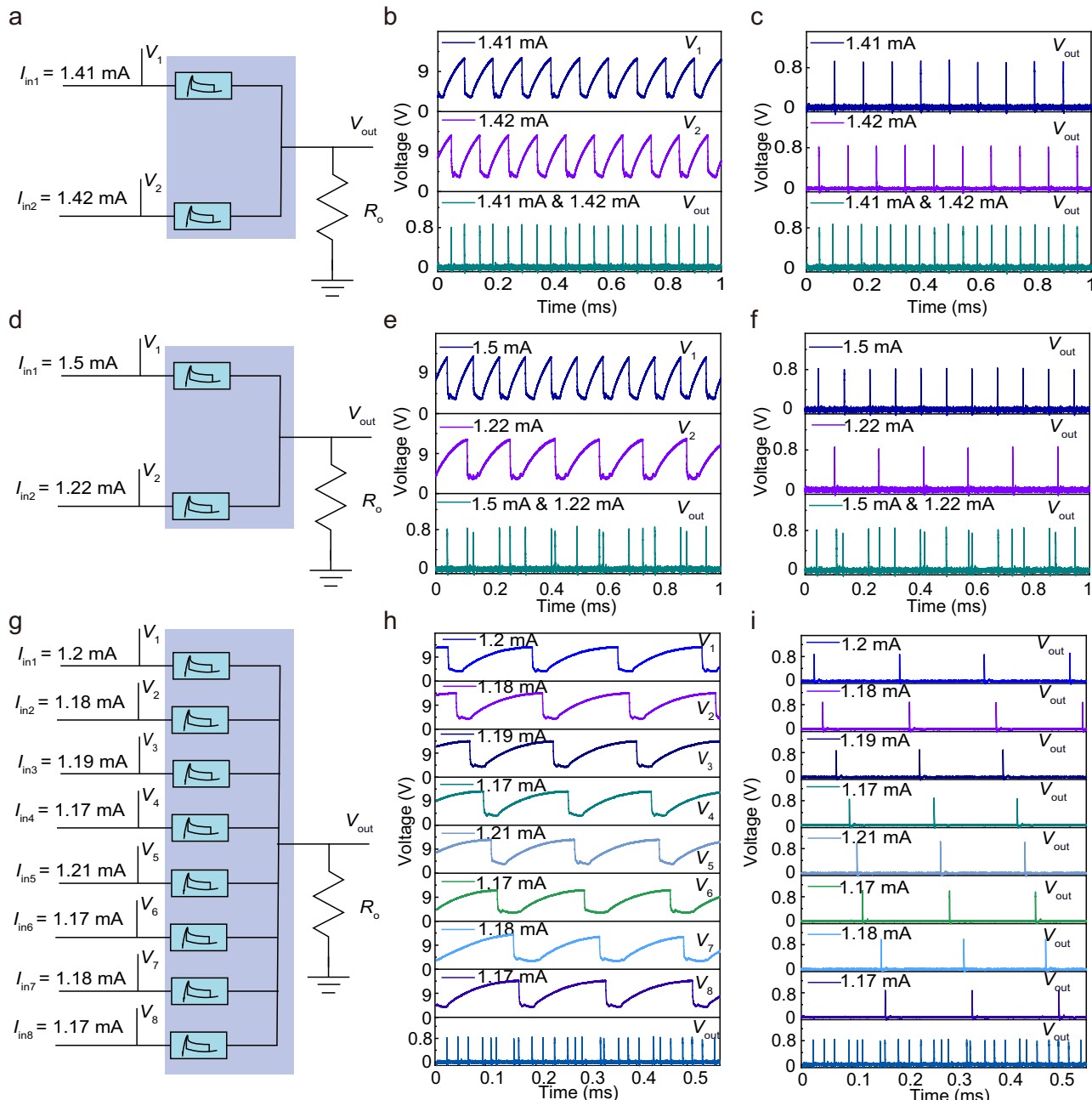

**Fig. 4 | Frequency converter using VO$_2$ memristor array-based oscillators.**
**a** Schematic diagram of the integration of pulse signals accumulated at the same frequency through two oscillators. **b** The output results with two current biases (1.41 mA and 1.42 mA) applied simultaneously. **c** The response $V_{out}$ with only one current bias 1.41 mA (first panel) or 1.42 mA (second panel) applied. The response $V_{out}$ with the same current biases applied simultaneously (third panel). **d** Schematic diagram of the accumulation of different frequencies through the integration of pulse signals from two oscillators. **e** The output results with two current biases (1.5 mA and 1.22 mA) applied simultaneously. **f** The response $V_{out}$ with only one current bias 1.5 mA (first panel) or 1.22 mA (second panel) applied. The response $V_{out}$ with the same current biases applied simultaneously (third panel). **g** Schematic diagram of frequency accumulation through pulse signal integration of 8 oscillators. **h** The output results with 8 current biases (1.2 mA, 1.18 mA, 1.19 mA, 1.17 mA, 1.21 mA, 1.17 mA, 1.18 mA and 1.17 mA) applied simultaneously. **i** The response $V_{out}$ with only one current bias 1.2 mA (first panel), 1.18 mA (second panel), 1.19 mA (third panel), 1.17 mA (fourth panel), 1.21 mA (fifth panel), 1.17 mA (six panel), 1.18 mA (seventh panel) or 1.17 mA (eighth panel) applied. The response $V_{out}$ with the same current biases applied simultaneously (ninth panel).

We compare the system performance with conventional CMOS-based frequency converter designs using this end-to-end hardware and software co-designed WIoT setup. For acoustic testing, we use a sensor audio stream source of 52.82 kB, with a duration of 3 s and a sampling frequency of 4.41 kHz (Fig. 5c). As shown in Fig. 5d, for vision testing, we use a sensor RGB source image of 300 × 300 pixels, each pixel is represented by three 8-bit unsigned numbers for red, green and blue components, respectively; hence the total number of bits that needs to be transmitted is 270 kB. For spatial data transmission in WIoT (Fig. 5e), we transmit point cloud data with XYZ coordinates (176.946 kB).

To verify the functionality of our VO$_2$-based frequency converter in the system, we first compare the transmission bit error rate (BER) with reference CMOS-based frequency converter. As shown in Fig. 5f, across different SNR choices, our memristive system introduces less than 0.21 dB, 0.13 dB and 0.02 dB BER degradation for point cloud, vision and acoustic data transmission, respectively, demonstrating correct functionality when applying it in real-world scenarios.

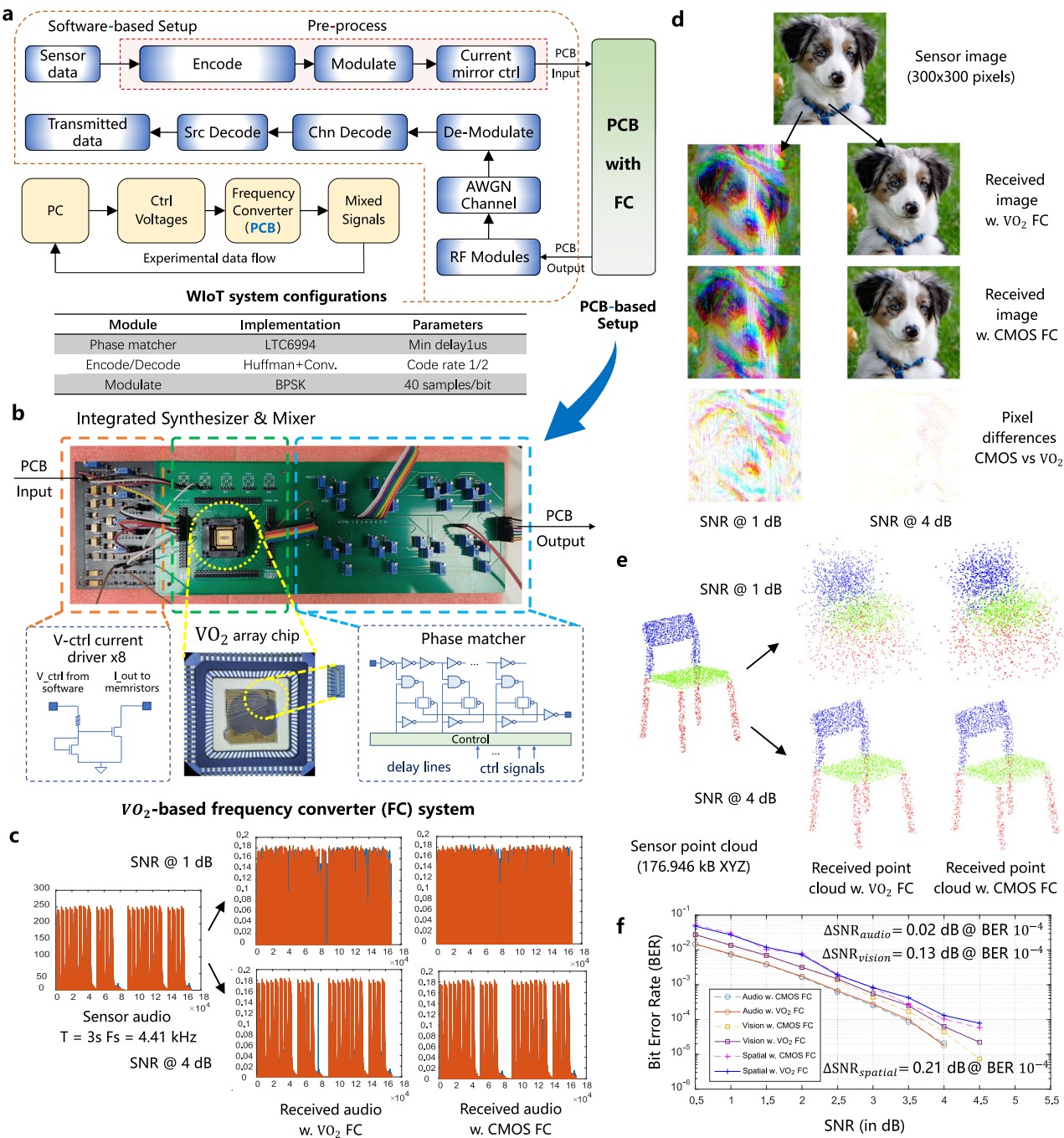

**Fig. 5 | Hardware and software co-designed experimental system using VO₂-based memristive frequency converter. a** The sensor data are captured using IoT vision, acoustic and depth sensors and the pre-processing are carried out in software to generate control voltages for voltage-controlled current drivers on testing PCB. **b** PCB board for the VO₂ memristor-based frequency converter with voltage-controlled current drivers as well as phase matchers to align with follow-up radio frequency modules. **c** Data transmission results under SNR 1 dB and 4 dB for sensor audio of 3 s with sampling frequency of 4.41 kHz. **d** Data transmission results under SNR 1 dB and 4 dB for sensor image of 300 × 300 pixels. **e** Data transmission results under SNR 1 dB and 4 dB for sensor point cloud of 176.946 kB of XYZ coordinates. **f** The BER performance using conventional CMOS-based frequency converter versus VO₂ memristor-based frequency converter.

Figure 5c–e present the raw audio, image and point cloud samples as well as the received samples using CMOS-based FC and VO₂ memristor-based FC at high (4 dB) and low (1 dB) SNR regimes. In high SNR regime (4 dB), our memristor-based FC demonstrates negligible transmission loss compared to CMOS-based FC and can efficiently recover the original datastreams from channel noise. In low SNR regime (1 dB) where noise level is high in transmission channel, the differences in received signals between our memristor-based FC and CMOS-based FC become more significant, but our VO₂-based FC can still maintain good synthesize and mixing functionalities.

Supplementary Figs. 23–26 show detailed transmission results using end-to-end WIoT experimental setup for a short-piece experimental source datastreams, vision datastreams, acoustic datastreams and spatial datastreams across different SNRs. Area cost estimation of VO₂-based FC using a 180 nm technology are given in Supplementary Table 2. Supplementary Table 3 shows that memristor-based FC outperforms conventional CMOS-based FC designs by more than 1.45×∼1.94× in power consumption, demonstrating promising prospect for future frequency converter design in wireless IoT systems at low frequency bands.

## Discussions

In this work, we design and fabricate $VO_2$ memristor array to enable a frequency converter that can executes in-situ frequency synthesize and mix functions for next-generation WIoT networks. We develop a highly uniform and epitaxially grown $VO_2$ memristor, based on which we build a calibratable $8 \times 8$ $VO_2$ memristor array with calibration resistors and capacitors as well as compact periphery circuits. We study the $VO_2$ array programmability in detail with different current drives and calibration resistances for synthesizing and mixing 2 ~ 8 channels with frequencies up to 48 kHz. We build an end-to-end software and hardware co-designed WIoT demonstration setup using $VO_2$-based frequency converter. When transmitting three representative sensor data (acoustic, vision and spatial) in WIoT experimental setup, our $VO_2$-based memristive frequency converter presents up to $1.45\times$ ~ $1.94\times$ power enhancement with less than 0.02 dB-0.21 dB BER performance degradations compared with conventional CMOS frequency converter. This work highlights the potential in using epitaxially grown $VO_2$ memristors and programmable $VO_2$-based memristive arrays for frequency convertor purpose, revealing a promising solution for next-generation WIoT system design at low frequency bands.

## Methods

### Fabrication of $VO_2$ memristor array

The 20 nm $VO_2$ films were epitaxially grown on $c$-$Al_2O_3$ substrates by employing the pulsed-laser deposition (PLD) approach with a 248 nm KrF excimer laser that was operated at an energy density of around $2\,J\,cm^{-2}$ and a repetition rate of 5 Hz. The $VO_2$ films were deposited at a temperature of 500 °C in an environment that contained oxygen that was moving and had a pressure of 1.0 Pa. After then, the temperature of the films was gradually decreased until they reached room temperature at a rate of $20\,°C\,min^{-1}$. The X-ray Reflection (XRR) technique was utilized so that the deposition rate of $VO_2$ thin films could be calibrated.

The $VO_2$ devices in this work were designed as a lateral structure with a width of 5 µm and channel length of 5 µm. Electron beam lithography (EBL), electron beam evaporation, and lift-off process were used to pattern the electrodes, which are composed of Au (50 nm) and Ti (5 nm) with a distance of 5 µm.

### Microstructural and Compositional Characterization

Raman spectra of $VO_2$ films grown on $c$-$Al_2O_3$ substrates were obtained using a Raman imaging microscope (Thermo Fisher Scientific DXRxi) with an excitation laser of 532 nm. The structure of $VO_2$ films was determined using XRD (D8 Discover, Bruker). X-ray photoelectron spectroscopy (XPS, Thermo Fisher Scientific-Escalab 250Xi) was used at ambient temperature to determine the elemental specific chemical state and stoichiometry of $VO_2$ films.

The cross-sectional TEM lamellar in this work were prepared by the focused ion beam (FIB, Helios G4, Thermo-fisher) technique using a dual-beam FIB system (FEI Helios Nanolab workstation) at 30 kV, with the final ion beam cleaning at 2 kV to minimize the surface amorphization. During the FIB patterning, the sample was initially coated with $SiO_2$ and a Pt layer, which was deposited using the electron beam to prevent surface damage. Following this, a higher-rate Pt coating was applied using a standard ion-beam method, which functioned as the bulk of the protective layer during the FIB cutting. TEM and STEM as well as EDS element mapping were performed at 200 kV (Talos 200FX, Thermo-fisher) equipped with Super-X SDD windowless EDS detector. High- resolution TEM and selected area electron diffraction results were analyzed by the Digital Micrograph software (Gatan Inc.). Scanning electron microscopy (SEM) was done on a field emission SEM (Merlin Compact).

### Electrical measurements

The $VO_2$ devices were placed in a MPI TS300-SE probe station in order to enable connections to source measurement unit, the external circuit and oscilloscope. As for measurements under different temperature in Supplementary Fig. 12, the $VO_2$ memristor was placed in a SUMMIT probe station. Electrical measurements were performed using an Agilent B1500A semiconductor parameter analyzer and KEYSIGHT DSOS404A digital storage oscilloscope. In Fig. 2d–f and Supplementary Fig. 5, Agilent B1500A perform continuous quasistatic DC current–voltage, and the the oscilloscope was used to measure the device's voltage. The experimental setup depicted in Supplementary Fig. 7 was used to connect the device to the external circuit for electrical measurements. In Figs. 3, 4 and Supplementary Figs. 8–22, Agilent B1500A was applied to create a constant DC current, and the oscilloscope was used to measure the voltage of the $VO_2$ devices and the resistance $R_0$.

### End-to-end wireless IoT experimental system design

The wireless IoT experimental setup shown in Fig. 5 is constructed following hardware and software co-design methodology. The sensor data is stored in a control PC where the pre-processor (including encode, modulate and current mirror control logic) are implemented in software. The encoding uses a classical probability-based Huffman code followed by rate-0.5 convolutional code with a generator matrix of [1 1 1;1 0 1]. The modulation block receives codewords from encoding block and employs binary phase shift keying (BPSK) at 40 samples/bit for constellation mapping and datastream generation. The frequency synthesize and mix using CMOS-based frequency converter utilize ideal sinusoidal waves at frequencies up to 48 kHz. On the other hand, the $VO_2$-based memristive frequency converter is implemented on the testing PCB as shown in Fig. 5 using fabricated $VO_2$ array chip. The PCB inputs (control voltages) are computed in control PC based on experimental data and are then sent to PCB to drive the on-board current drivers (built by CMOS devices) for in-situ synthesize and mix. To minimize phase mismatches of multiple frequency channels, we use programmable delay lines built by LTC6994 modules to calibrate the output phases of mixed signals before carrying out actual experiments for up to 8 mixing channels. The testing PCB outputs are collected using oscilloscopes and post-processed by control PC for end-to-end evaluations of WIoT applications (Supplementary Figs. 23–26).

## Data availability

All data supporting this study and its findings are available within the article, its Supplementary Information and associated files. The audio data is originally created in this work. The image data is originally from [https://www.npaws.org/]. The spatial data is originally from [https://github.com/charlesq34/pointnet2]. All source data for this study have been deposited in [https://doi.org/10.5281/zenodo.10477180] or are available from the corresponding author upon request.

## Code availability

The codes supporting the findings of this study are available from the corresponding authors upon request.

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

## Acknowledgements

This work was supported by the National Key R&D Program of China (2023YFB4502200, 2023YFA1407200), the National Natural Science Foundation of China (61925401, 92064004, 61927901, 8206100486, 92164302), Beijing Natural Science Foundation (L234026) and the 111 Project (B18001).

## Author contributions

C.L., Y.T., and Y.Y. designed the experiments. C.L., L.C., R.Y., and Y.T. fabricated the VO$_2$ devices. C.L., Y.T., W.Y., Z.P., and T.Z. performed electrical measurements. C.L., P.J.T., T.Z., Y.W., and Y.T. performed the simulations. C.L., Y.T., and Y.Y. prepared the manuscript. Y.T. and Y.Y. directed all the research. All authors analyzed the results and implications and commented on the manuscript at all stages.

## Competing interests

The authors declare no competing interests.
