## [Peer Review File · Nature Communications]

REVIEWER COMMENTS

Reviewer #1 (Remarks to the Author):

In this manuscript, Liu et al. fabricated VO₂ memristors and used them to construct frequency converter for wireless internet of things. VO₂ memristors are widely used as neurons in spiking neural networks or bio-inspired hardware, and in my opinion, this work goes beyond the state-of-the-art and uses VO₂ memristor array for frequency converter with in-situ frequency synthesizer and mix for the first time, which has not yet been reported. This paper develops the programming methodology by tuning the calibration resistance connected to the VO₂ array, followed by shaping and phase locking circuits to match with existing communication link, showing better performance than conventional frequency converter with separate frequency synthesizer and mix. The in-situ frequency synthesizer and mix enabled by VO₂ memristor array are brand new.

In fact, the study is very complete by providing details on materials synthesis, VO₂ device fabrication, SPICE modelling of VO₂ device, VO₂ array characterization, and end-to-end system implementation/evaluation under different SNRs in a real-world wireless internet of things link. Moreover, the performance of the VO₂ frequency converter outperforms the state-of-the-art. The texts and figures of this paper are clear. With some minor revisions, the paper will be very good for publication. Here are a few minor questions that I think need to be clarified before the publication of this manuscript:

- 1) Frequency converter in communication systems are running all the time whenever there are data transmission in real-world applications. Did the authors study the endurance of such VO₂ devices under current drives?
- 2) From the micrograph we can optically see the conductive filament of the VO₂ device, which have been reported in similar planar VO₂ devices. But the frequency that this device can support looks low. Did the authors connect the device with a capacitance? It is better to explain more details.
- 3) Does the temperature affect the frequency converter performance? In real-world wireless internet of things applications, the temperature may change with environment. How the VO₂ memristor behaves under varying temperature?
- 4) The end-to-end system implementation is divided into hardware PCB and software. How the datastreams are communicated between PCB and software are not very clear. It is better to provide more details on the end-to-end experimental setup.

Other than these questions, this study deserves publication in Nature Communications.

Reviewer #2 (Remarks to the Author):

The author presented the fabrication of VO₂ memristor arrays and demonstrated frequency converter for wireless IOT applications. By going through the manuscript, I have an impression that the article is not suitable for publication in Nature Communications and may be better suited for specialized journal with focus on electronic devices. Some of my specific comments are as follows.

1. From the abstract to the end, the focus is given to device application with emphasis on frequency converter with the mention of IOT, Data transmission etc., without any clear scientific advancement or novelty.
2. While the device fabrication and the mentioned frequency converter detail may be suitable for more specialized journals, it is difficult to follow up for general scientific community.
3. In terms of data analysis and interpretation, more detailed characterizations are needed to support the phase transition characteristics of VO₂. For example, did the author measure the R-T plot to show the insulator-metal transition quality? Further, XRD, Raman and XPS analysis are needed.
4. Regarding the TEM lattice image indexing, author confirmed the tetragonal phase of VO₂ which is incorrect. At room temperature, VO₂ supposed to show monoclinic phase and not tetragonal.

Reviewer #1 (Remarks to the Author):

In this manuscript, Liu et al. fabricated VO2 memristors and used them to construct frequency converter for wireless internet of things. VO2 memristors are widely used as neurons in spiking neural networks or bio-inspired hardware, and in my opinion, this work goes beyond the state-of-the-art and uses VO2 memristor array for frequency converter with in-situ frequency synthesizer and mixer for the first time, which has not yet been reported. This paper develops the programming methodology by tuning the calibration resistance connected to the VO2 array, followed by shaping and phase locking circuits to match with existing communication link, showing better performance than conventional frequency converter with separate frequency synthesizer and mixer. The in-situ frequency synthesizer and mixer enabled by VO2 memristor array are brand new.

In fact, the study is very complete by providing details on materials synthesis, VO2 device fabrication, SPICE modelling of VO2 device, VO2 array characterization, and end-to-end system implementation/evaluation under different SNRs in a real-world wireless internet of things link. Moreover, the performance of the VO2 frequency converter outperforms the state-of-the-art. The texts and figures of this paper are clear. With some minor revisions, the paper will be very good for publication. Here are a few minor questions that I think need to be clarified before the publication of this manuscript:

Our response: We express our sincere gratitude to the reviewer for the positive comments. The reviewer's comments and suggestions have been highly valued, and in response, we have conducted more experiments (Supplementary Fig. 11, 12, 27, 28). The point-by-point responses to each question and corresponding changes are presented below.

1) Frequency converter in communication systems are running all the time whenever there are data transmission in real-world applications. Did the authors study the endurance of such VO2 devices under current drives?

Our response: We would like to thank the reviewer for the constructive suggestion.

The endurance of the VO₂ memristor has to be evaluated to ensure the dependability of frequency converter. Thus, we characterized the endurance of the VO₂ memristor by connecting it in an oscillation circuit (Fig. 3a) under 2.1 mA applied current with a width of 66 s. As evident in Supplementary Fig. 11, the VO₂ memristor can withstand >10⁶ switching cycles without any noticeable degradation in the oscillating behavior, demonstrating very high endurance.

We have added the new experimental results in Supplementary Fig. 11, and we also added the following discussion into the revised manuscript:

- Page 16:

Significantly, the VO₂ oscillator exhibited a remarkable endurance of >10⁶ switching cycles, as depicted in Supplementary Fig. 11. This characteristic guarantees the dependability of our frequency converter that incorporate these particular devices.

Supplementary Figure 11. Endurance of the VO₂ oscillator. Experiment results of oscillator under 2.1 mA applied current with a width of 66 s.

2) From the micrograph we can optically see the conductive filament of the VO₂ device, which have been reported in similar planar VO₂ devices. But the frequency that this device can support looks low. Did the authors connect the device with a capacitance? It is better to explain more details.

Our response: We would like to sincerely thank the reviewer for the detailed suggestions. To address this question, we have added the following discussion in the page 12 of revised manuscript: “The simple circuit used to obtain the optical micrograph video is shown in Fig. 3a, with the parallel capacitance and current bias being 110 μ F and 1.6 mA, respectively. The lower panel in the video is the optical micrograph video, and the upper panel is the generated spike signal V_{mem} .”

3) Does the temperature affect the frequency converter performance? In real-world wireless internet of things applications, the temperature may change with environment. How the VO₂ memristor behaves under varying temperature?

Our response: We would like to thank the reviewer for the valuable suggestion.

The temperature may vary depending on the setting in wireless Internet of Things applications. Therefore, determining the correlation between the frequency of the VO₂ oscillator and temperature is crucial. In light of this, we have performed new studies to measure the characteristics of VO₂ devices under different temperatures (Supplementary Fig. 1, 12), and have added relevant discussions in the revised manuscript:

- Page 9-10:

As shown in Supplementary Fig. 1a, the quality of the VO₂ film was confirmed electrically through isothermal electrical resistance characterization of the metal-insulator transition (MIT) in lateral device. The MIT shows repeatable resistance switching with a magnitude of $\approx 400\times$. Supplementary Fig. 1b further confirms critical temperatures of about 345 and 339 K for the heating and cooling processes, respectively. The large change in resistance during switching and the measured thermal hysteresis are similar to values found in the literature for undoped VO₂ (Ref. R1).

- Page 16:

The temperature may vary depending on the setting in wireless Internet of Things applications. Therefore, determining the correlation between the frequency of the VO₂ oscillator and temperature is crucial. During the temperature-elevated test, only the VO₂ device was heated in the probe station, while other components were kept at room temperature. The experimental results (Supplementary Fig. 12) demonstrate that the frequency of the VO₂ oscillator rises with increasing temperature. This should be taken into account using temperature-dependent oscillation models such as the one described in ref. 42 when deploying our frequency converter in real-world applications.

Supplementary Figure 1. Characterization of VO₂ memristor. (a) Temperature-dependent resistance switching plot and the corresponding (b) differential curve of the VO₂ film. The heating and cooling branches are represented by the red and blue curves, respectively. (c) Comparison of XRD pattern before and after VO₂ film deposition. (d) Comparison of Raman spectra before and after VO₂ film deposition.

Supplementary Figure 12. Oscillation frequency response under different temperature. As the temperature increases, the oscillation frequency increases.

4) The end-to-end system implementation is divided into hardware PCB and software. How the datastreams are communicated between PCB and software are not very clear. It is better to provide more details on the end-to-end experimental setup.

Our response: We would like to thank the reviewer for the valuable suggestion.

We edit Supplementary Figure 23, Supplementary Table 3 and added Supplementary Figure 27, 28 and Supplementary Table 2 with relevant contents to illustrate the details

of the end-to-end experimental setup and also show how datastreams are communicated between PCB and software.

Supplementary Figure 23. Measurements of frequency synthesize and mix using CMOS-based frequency converter versus VO₂ memristor-based frequency converter on an experimental source datastreams from WIoT sensors. In this example, VO₂ memristor-based frequency converter reaches the same performance of CMOS-based frequency converter in SNR 1.5dB.

Supplementary Figure 27. Detailed illustration of each module in end-to-end wireless IoT demonstration system and interface between PCB and software.

Step ①: Sensor data are: 1) audio datastream of 3 seconds with sampling frequency of 4.41KHz (each sampling point is quantized using FP32 and the total data size is 52.92KB); 2) image datastream of $300p \times 300p$ (each pixel is quantized as INT8 RGB and the total data size is 270KB); 3) point cloud datastream (with data size 176.946KB). These data are sent to Encode block as a 1-dimensional bitstream and its length is determined by the sensor data size.

Step ② : The Encode block includes: 1) Huffman encoding given a vector of probability of the unique symbols. The probability is computed based on symbol count divided by the total length of pre-transmitted datastream. We use the built-in function `huffmanenco` in MATLAB for this process; 2) Convolutional coding for channel encoding, which uses a generator matrix of $[1 \ 1 \ 1; 1 \ 0 \ 1]$ with code rate $1/2$. The block diagram of the convolutional encoder is shown in Supplementary Figure 28. Depending on the number of frequency channels, the resulting datastream from Encode block can remain as two 1-dimensional encoded bitstreams or demultiplexed into 8 1-dimensional encoded bitstreams.

Supplementary Figure 28. Detailed illustration of encoding/modulate blocks and interfaces with PC.

Step ③: The Modulate block receives the 2~8 1-dimensional bitstreams from Encode block and uses binary phase shift keying (BPSK) to map the input bit 0's or 1's. The 0's and 1's in the bitstreams are mapped to -1 and +1. These -1's and +1's are sent to current mirror ctrl block to generate physical voltages for mixing purpose.

Step ④: The Current mirror ctrl block is still in software which receives mapped bitstreams from Modulate block and generates current mirror ctrl voltages based on Supplementary Figure 14, 15, 16 and 22. The generated ctrl voltages are stored and used to configure parallel voltage-controlled current sources from 0~2mA to drive the VO₂ memristors for follow-up frequency synthesise and mix on PCB.

Step ⑤: We use a delay-line based phase matcher to first manually adjust the phases of the VO₂ generated oscillation waves on PCB. Upon completion of the phase matching, we drive the VO₂ based FC using current sources configured by pre-stored

ctrl voltages on PCB block. The in-situ synthesized and mixed analog signals are stored with oscilloscopes.

Step ⑥: The mixed analog signals are sent back to the software where the radio frequency block using 915 MHz for short-range connectivity is employed.

Step c: The AWGN block adds white Gaussian noise to the RF signals based on set SNR levels. The addition uses MATLAB awgn function.

Step ⑧: The Demodulate block receives noisy analog streams from AWGN block and map -1's and +1's back to 0's and 1's, respectively.

Step ⑨: The channel decoder block decodes the received bitstream using Viterbi decoder. The Viterbi decoder trellis is pre-determined by generator matrix G in step 2 and we use MATLAB vitdec function for this purpose.

Step i: The src decoder block receives the channel decoded bitstream and decode the source bitstream based on Huffman coding scheme.

Supplementary Table 2. Area Cost Estimation of VO₂-based frequency converter using 180 nm process technology.

Circuit Module (180nm)	Number	Area (mm ²)
VO ₂ Driver	8	0.11
VO ₂ Memristor Array	8×8	0.16
Delay Line	8	0.038
DeMux	2	0.013
Periphery Circuits (Cap/Res/...)	8	1.62

Total	N/A	1.941
-------	-----	-------

Supplementary Table 3. Comparison with CMOS-based frequency converter

Frequency Converter	CMOS-based ¹ (Qorvo)	CMOS-based ⁶ (Analog Devices)	VO ₂ Memristor-based (This work)	
Technology	40nm	65nm	180nm	
Main Applications	Radio-freq./WIoT	Radio-freq./WIoT	Energy-efficient WIoT	
Frequency Range	30-2500MHz	1KHz-3000MHz	Up to 48KHz	
Latency	> 5ns	3s	-50ns	
Area Cost	5mm × 5mm (QFN)	4mm × 4mm (QFN)	1.941mm ²	
Synthesize & Mix	Both	Mix Only	Both	
Perf. (Acoustic/Vision/Spatial)	BER	3.47dB/3.71dB /4.03dB	3.47dB/3.71dB /4.03dB	3.49dB/3.88dB /4.24dB
	Power	165-225mW	126-147mW	85.2-114.1mW

The reference CMOS-based frequency converters for comparison are implemented using more advanced process technology (from Qorvo¹ and Analog Devices⁶).

Other than these questions, this study deserves publication in Nature Communications.

Our response: We would like to thank the reviewer once again for the positive assessment.

Reviewer #2 (Remarks to the Author):

The author presented the fabrication of VO₂ memristor arrays and demonstrated frequency converter for wireless IOT applications. By going through the manuscript, I have an impression that the article is not suitable for publication in Nature Communications and may be better suited for specialized journal with focus on electronic devices. Some of my specific comments are as follows.

Our response: We would like to sincerely appreciate the reviewer's constructive feedbacks, which are extremely valuable in improving the quality of our work. We have revised the manuscript to emphasize the novelty and scientific advancements which are also presented in responses to reviewer's specific comments. Please kindly check. We also summarize the key novelty of our work here:

- 1) For the first time, for efficient frequency converter design purpose, we did comprehensive study on self-oscillation phenomenon of VO₂ under different circumstances and optimized the cycle-to-cycle and device-to-device uniformity;
- 2) For the first time, frequency converter is designed utilizing an array of memristive oscillators that enables in-situ frequency synthesize and mixing operations: We comprehensively study the programmability of calibratable VO₂ memristor array and the accuracy of such frequency converter. We demonstrate that this approach holds significant promise for enhancing the latency and energy efficiency of WIoT hardware systems.
- 3) For the first time, we build an end-to-end software-hardware co-designed WIoT

experimental system to evaluate the VO₂ memristive frequency converter for real-world applications: Instead of demonstrating single-memristor device or single-array performance, we comprehensively evaluate the end-to-end performance of our frequency converter design in a WIoT link.

1. From the abstract to the end, the focus is given to device application with emphasis on frequency converter with the mention of IOT, Data transmission etc., without any clear scientific advancement or novelty.

Our response: We would like to sincerely thank the reviewer for this advice and clarify the reviewer's concerns as follows. The manuscript is also revised accordingly. The scientific advancement or novelty are summarized as below:

1) Highly uniform VO₂ memristor device for frequency converter: VO₂ memristor exhibits self-oscillation phenomenon based on negative differential resistance (NDR) behavior in the current-driven *I-V* curves. Optically visible filaments recorded in-situ show that self-oscillation within the NDR regions is due to the formation and disappearance of a high temperature conducting channel, which is a result of electrothermally-induced Mott transition. Thus, for the first time, a highly uniform, calibratable oscillator based on VO₂ memristor has been experimentally implemented with help of compact circuits design. The oscillator has excellent cycle-to-cycle and device-to-device uniformity, due to the high crystalline quality of epitaxially grown VO₂ and introduction of parallel calibration resistor. We did comprehensive study on VO₂ memristor aiming to support follow-up frequency converter design.

2) VO₂ memristor enabled, programmable frequency converter with in-situ frequency synthesize and mix: For the first time, frequency converter is designed utilizing an array of memristor oscillators that enables the integration of frequency synthesize and mixing operations inside a single module. This approach holds significant promise for enhancing the speed and energy efficiency of WIoT hardware systems. We build an 8x8 VO₂ memristor array along with its surrounding peripheral circuitry. The surrounding circuits are compact with minimal area cost overheads. We conduct an in-depth analysis of the programmability of the VO₂ array using a variety of current drivers in order to synthesize and mix up to eight datastreams at frequencies of up to about 48 KHz. The in-situ synthesize and mix functions within a single module has not been reported to date.

4) End-to-end performance evaluation of VO₂ memristor enabled frequency converter: Rather than exploiting the new VO₂ memristor based frequency converter based only on single-device or array testing, we build an end-to-end software-hardware co-designed WIoT experimental system to study the memristive frequency converter for real-world applications. This is the first time a VO₂ based frequency converter is evaluated in a detailed end-to-end system experimental setup. We carry out comprehensive experiments with representative sensors including acoustic, vision and spatial data. The integration for an end-to-end demonstration system is novel and demonstrates great potential for real-world adoptions for next-generation WIoT systems.

2. While the device fabrication and the mentioned frequency converter detail may be suitable for more specialized journals, it is difficult to follow up for general scientific community.

Our response: We would like to sincerely thank the reviewer for this advice. As mentioned above, this work highlights the first reported VO₂ memristor-based frequency converter and shows VO₂ memristor's great potential in solving frequency conversion's speed and energy efficiency problems using high crystalline quality epitaxially grown VO₂ and a newly designed calibratable VO₂-based memristive oscillation array with compact circuits design, revealing a promising direction for next-generation WIoT system design. We believe the three novelty will help the general scientific community to better understand how frequency converter can be realized using VO₂ memristors.

3. In terms of data analysis and interpretation, more detailed characterizations are needed to support the phase transition characteristics of VO₂. For example, did the author measure the R-T plot to show the insulator-metal transition quality? Further, XRD, Raman and XPS analysis are needed.

Our response: We would like to sincerely thank the reviewer for this very helpful advice.

The quality of the VO₂ film was confirmed electrically through isothermal electrical resistance characterization (*R-T*) of the metal-insulator transition (MIT) in lateral device, as shown in Supplementary Fig. 1a. The MIT shows repeatable resistance switching with a magnitude of $\approx 400\times$. Supplementary Fig. 1b further confirms critical

temperatures of about 345 and 339 K for the heating and cooling processes, respectively. The large change in resistance during switching and the measured thermal hysteresis are similar to values found in the literature for undoped VO₂ (Ref. R1), verifying the high crystalline quality of the epitaxially grown VO₂. Not only that, we have determined the correlation between frequency of the VO₂ oscillator and temperature. During the temperature-elevated test, only the VO₂ device was heated in the probe station, while other components were kept at room temperature. The experimental results (Supplementary Fig. 12) demonstrate that the frequency of the VO₂ oscillator rises with increasing temperature. Besides, the VO₂ crystal quality were also characterized by XRD, Raman spectroscopy and XPS, once again verifying the high crystalline quality of the VO₂ film.

To address this question, we have added the new experimental results in Supplementary Fig. 1, 2, 12 and added the following discussion into the revised manuscript:

- Page 9-11:

As shown in Supplementary Fig. 1a, the quality of the VO₂ film was confirmed electrically through isothermal electrical resistance characterization of the metal-insulator transition (MIT) in lateral device. The MIT shows repeatable resistance switching with a magnitude of $\approx 400\times$. Supplementary Fig. 1b further confirms the critical temperatures of about 345 and 339 K for the heating and cooling processes, respectively. The large change in resistance during switching and the measured thermal hysteresis are similar to values found in the literature for undoped VO₂ (Ref. R1). The

VO₂ crystal quality were also characterized by X-ray diffraction (XRD), Raman spectroscopy, X-ray photoelectron spectroscopy (XPS), and scanning transmission electron microscope (STEM). The results of the XRD taken both before and after the VO₂ film was deposited are depicted in Supplementary Fig. 1c. According to the findings of the XRD analysis, there is a single diffraction peak at 39.8 degrees occurring in the film, which represents (020) monoclinic VO₂ films formed on *c*-plane Al₂O₃ substrates (Ref. R2). The Raman measurement (Supplementary Fig. 1d) showed that all the peaks are in good agreement with reported monoclinic VO₂ results (Ref. R3). The chemical composition and valence state of the VO₂ microcrystals were investigated using XPS. The confirmation of the presence of elements C, O, and V was achieved through the observation of their respective XPS characteristic peaks, as depicted in Supplementary Fig. 2a. The observed peaks for element C may potentially be attributed to the presence of carbon on the surface of the samples, as depicted in Supplementary Fig. 2b. The O 1s peak (Supplementary Fig. 2c) can be fitted with three peaks at 530.50 eV (V-O bond), 532 eV (O-H bond resulting from physical absorption of H₂O on the surface of the sample) and 532.67 eV (C-OH bond originating from organic hydroxyl) that all correspond to the oxygen valence of O²⁻. The high-resolution XPS spectrum of the V 2p^{3/2} (Supplementary Fig. 2d) shows that clear peaks can be deconvoluted into V₄₊ and V⁵⁺ states centered at 516.21 eV and 517.35 eV (Ref. R4), respectively. Only the first 1-3 nm of the sample volume are detected by XPS, which means that surface oxidation is primarily responsible for the large shoulder linked to V⁵⁺.

The temperature may vary depending on the setting in wireless Internet of Things applications. Therefore, determining the correlation between frequency of the VO₂ oscillator and temperature is crucial. During the temperature-elevated test, only the VO₂ device was heated in the probe station, while other components were kept at room temperature. The experimental results (Supplementary Fig. 12) demonstrate that the frequency of the VO₂ oscillator rises with increasing temperature. This should be taken into account using temperature-dependent oscillation models such as the one described in ref. 42 when deploying our frequency converter in real-world applications.

Supplementary Figure 1. Characterization of VO₂ memristor. (a) Temperature-dependent resistance switching plot and the corresponding (b) differential curve of the VO₂ film. The heating and cooling branches are represented by the red and blue curves, respectively. (c) Comparison of XRD pattern before and after VO₂ film deposition. (d) Comparison of Raman spectra before and after VO₂ film deposition.

Supplementary Figure 2. X-ray photoelectron spectrum (XPS) analysis. (a) XPS spectra survey and high-resolution (b) C 1s, (c) O 1s and (d) V 2p core level spectra for VO₂ film.

Supplementary Figure 12. Oscillation frequency response under different temperature. As the temperature increases, the oscillation frequency increases.

4. Regarding the TEM lattice image indexing, author confirmed the tetragonal phase of VO₂ which is incorrect. At room temperature, VO₂ supposed to show monoclinic phase and not tetragonal.

Our response: We would like to sincerely thank the reviewer for pointing out the mistake. We have now corrected it in the revised manuscript.

References

- R1 Taha, M. et al. Insulator-metal transition in substrate-independent VO₂ thin film for phase-change devices. *Sci. Rep.* **7**, 17899 (2017).
- R2 Choi, Y. et al. Correlation between symmetry and phase transition temperature of VO₂ films deposited on Al₂O₃ substrates with various orientations. *Adv. Electron. Mater.* **7**, 200874 (2021).
- R3 Schilbe, P. Raman scattering in VO₂. *Physica B Condens. Matter.* **316**, 600–602 (2002).
- R4 Brown, T. D. et al. Electro-thermal characterization of dynamical VO₂ memristors via Local activity modeling. *Adv. Mater.* 2205451 (2022).

REVIEWERS' COMMENTS

Reviewer #1 (Remarks to the Author):

I thank the authors for the prompt responses which have well addressed my early concerns. I thus recommend it for publication as is.

Reviewer #2 (Remarks to the Author):

The revised version and responses from author addressed my comments. For example, the scientific part of the device performance is described with more details now. In addition, the author also carried out additional studies and strengthened materials characterization part with the use of Raman, XPS and R-T plots. Overall, the revised manuscript shows a good balance between scientific findings and technological applications.

This may be considered for the publication in Nature Communications.